# 3D genomic analysis reveals novel enhancer-hijacking caused by complex structural alterations that drive oncogene overexpression

Katelyn L. Mortenson [1], Courtney Dawes[2], Emily R. Wilson[1], Nathan E. Patchen[2], Hailey E. Johnson [1,3], Jason Gertz[1], Sweke D. Bailey [4,5], Yang Liu[2], Katherine E. Varley[1] ✉ & Xiaoyang Zhang [1] ✉

Cancer genomes are composed of many complex structural alterations on chromosomes and extrachromosomal DNA (ecDNA), making it difficult to identify non-coding enhancer regions that are hijacked to activate oncogene expression. Here, we describe a 3D genomics-based analysis called HAPI (Highly Active Promoter Interactions) to characterize enhancer hijacking. HAPI analysis of HiChIP data from 34 cancer cell lines identified enhancer hijacking events that activate both known and potentially novel oncogenes such as *MYC, CCND1, ETV1, CRKL*, and *ID4*. Furthermore, we found enhancer hijacking among multiple oncogenes from different chromosomes, often including *MYC*, on the same complex amplicons such as ecDNA. We characterized a *MYC-ERBB2* chimeric ecDNA, in which *ERBB2* heavily hijacks *MYC*'s enhancers. Notably, CRISPRi of the *MYC* promoter led to increased interaction of *ERBB2* with *MYC* enhancers and elevated *ERBB2* expression. Our HAPI analysis tool provides a robust strategy to detect enhancer hijacking and reveals novel insights into oncogene activation.

Somatic structural variants, including deletions, duplications, inversions, and translocations, dramatically alter genomic structures in cancer cells. These events can be readily detected by DNA imaging, whole-genome/long-read sequencing, or Hi-C-based 3D genomic assays[1–5]. Understanding the functional consequences of structural variants is key to uncovering the mechanisms of tumorigenesis and developing therapeutic strategies. For instance, structural variants can create a fusion gene with oncogenic properties such as *BCR-ABL1* and *EML4-ALK*[6,7]. However, most of the breakpoints of cancer-associated structural variants fall in noncoding regions[8]. These variants can place enhancer elements adjacent to oncogenes such as *MYC, MYCN, MYB*,

*CCND1, EVI1*, and *GFI1*, and activate their expression, events known as enhancer hijacking[9–17].

Several analysis approaches have been developed to identify enhancer hijacking. PANGEA and CESAM identify candidate genes based on the correlation of their expression with adjacent structural alterations across tumor samples, which require large cohorts of data and do not utilize chromatin interaction evidence of enhancer hijacking[18,19]. Neoloop reconstructs Hi-C maps in structurally altered cancer genomes to identify aberrant chromatin loops and their associated genes[20], however, it does not prioritize the gene targets and the underlying functional enhancers. As structural variants often affect

[1]Department of Oncological Sciences, Huntsman Cancer Institute, University of Utah, Salt Lake City, UT, USA. [2]Department of Biochemistry, University of Utah, Salt Lake City, UT, USA. [3]Department of Cell Biology and Physiology, Brigham Young University, Provo, UT, USA. [4]Cancer Research Program, Research Institute of the McGill University Health Centre, Montreal, QC, Canada. [5]Department of Surgery and Human Genetics, McGill University, Montreal, QC, Canada. ✉e-mail: Katherine.Varley@hci.utah.edu; Xiaoyang.Zhang@hci.utah.edu

many enhancer-promoter interactions, most of which are likely passenger events[2,3], it is essential to prioritize the functional events. Furthermore, recent work has revealed that complex structural variants such as chromothripsis lead to the formation of circular extrachromosomal DNA (ecDNA)[21]. These events often cause extensive alterations of enhancer-promoter interactions, posing another challenge to characterize the underlying mechanisms of oncogene regulation[22–24].

Here, we propose a 3D genomics-based strategy to identify and characterize enhancer hijacking events based on two assumptions: 1) oncogenes subject to enhancer hijacking should be highly regulated by enhancers; 2) the hijacked enhancers should contribute a substantial proportion of an oncogene's enhancer usage. To test this, we utilized H3K27ac HiChIP data to map enhancer-promoter interactions and measured their interaction intensities. We applied our strategy to HiChIP data from 34 cancer and two normal cell lines. This approach identified known and novel enhancer hijacking events, prioritized their target genes, and defined the underlying enhancers. We also revealed novel enhancer hijacking events associated with ecDNAs or complex chromosomal amplicons. In these events, multiple oncogenes from different chromosomal regions are translocated near each other, amplified together, and hijack each other's enhancers to boost their overexpression.

## Results

### HAPI analysis identifies genes that are highly interactive with enhancers

We first developed an analysis method, Highly Active Promoter Interactions (HAPI), that utilizes H3K27ac HiChIP data to quantify all enhancer interactions of each gene's promoter in the genome, including intra- and inter-chromosomal interactions. This method ranks genes by 1) their enhancer contact value defined as the number of interacting enhancers and 2) their interaction intensity as defined by the number of HiChIP paired-end tags (PETs). We define highly interactive (HAPI) genes as those that meet or exceed the inflection points of both the ranked enhancer contact value and ranked interaction intensity in a manner similar to previously published methods that define super-enhancers[25] (Fig. 1A), as exemplified in the prostate cancer cell line LNCaP (Fig. 1B and S1A, B).

We applied HAPI analysis to H3K27ac HiChIP data that we and others generated in 34 cancer cell lines representing multiple cancer types (Supplementary Table 1), which identified on average 538 HAPI genes in each cell line (Supplementary Data 1). In LNCaP cells, HAPI genes are more likely to interact with super-enhancers compared to non-HAPI enhancer-connected genes as expected[26,27] (Fig. S1C), but there are additional highly interactive HAPI genes that do not utilize super-enhancers (Figure S1C). Conversely, super-enhancers are not

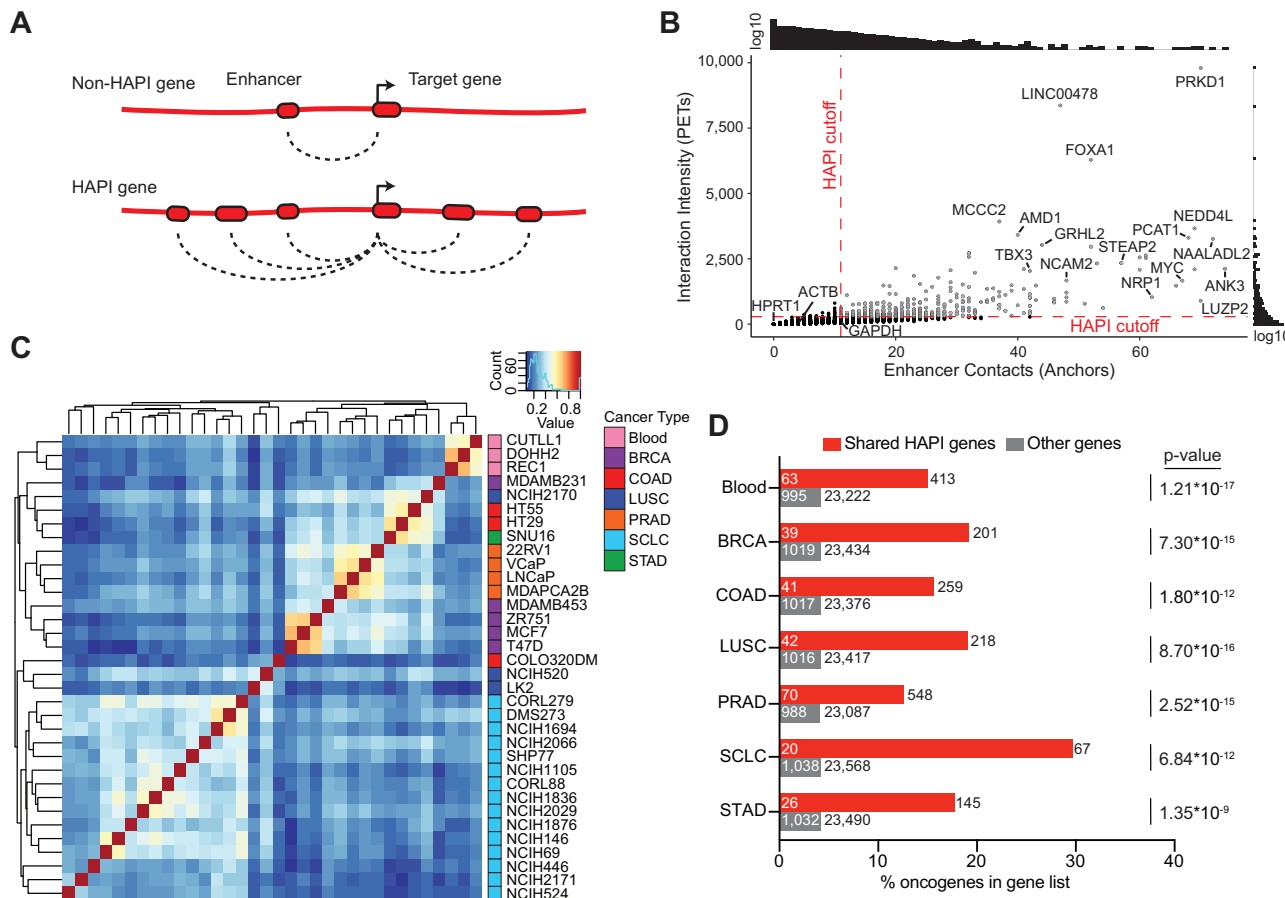

**Fig. 1 | HAPI analysis identifies cancer-related genes that are highly interactive with enhancers. A** Schematic models illustrating the identification of HAPI genes based on enhancer interactions. **B** Plotted are the number of enhancers (x-axis) and interaction intensity (y-axis) for each gene in LNCaP cells. HAPI cutoff is defined by the inflection point of each scoring metric. **C** Spearman correlation clustering of cancer cell lines based on interaction scores of their associated HAPI genes. BRCA breast cancer, COAD colorectal adenocarcinoma, LUSC lung squamous cell carcinoma, PRAD prostate adenocarcinoma, SCLC small cell lung cancer, STAD stomach adenocarcinoma. **D** The percentage of HAPI genes (shared by at least half of the cell lines in each cancer type) that are previously annotated as oncogenes. The numbers outside and within each bar indicate the total number of shared HAPI (red) or other (gray) genes and the ones annotated as oncogenes, respectively. *P* values are derived from two-sided Fisher's exact tests.

necessarily looped to HAPI genes in LNCaP cells, as most of the super enhancers interact with both HAPI and non-HAPI genes, or only non-HAPI genes (Fig. S1D). Overexpressed oncogenes such as *MYC*, *FOXA1*, *TBX3*, and *PCAT1* were identified as HAPI genes in LNCaP cells. Notably, HAPI genes are not necessarily genes with the highest expression, as highly expressed housekeeping genes *ACTB*, *HPRT1*, and *GAPDH* fell below the HAPI cutoff (Fig. 1B and S1E). Furthermore, when we investigated whether copy number amplification had an impact HAPI gene metrics, we found that only a small proportion of the amplified genes are identified as HAPI genes in LNCaP cells (Fig. S1F). We did not normalize copy number variants in this analysis, as we and others have shown that focal copy number amplifications contribute to enhancer-promoter interactions for oncogenes – a signal that we want to preserve in our analysis[28–33].

We performed unsupervised clustering of the cell lines based on a combination of the enhancer contact and interaction intensity values for each identified HAPI gene (Fig. 1C, see Methods for more details). The results showed that cell lines from the same lineages tend to cluster together, highlighting the cancer type-specificity of HAPI genes. We also found significant enrichments of previously annotated oncogenes[34–36], and genes involved in cancer-related pathways in our identified HAPI gene lists (Fig. 1D and Supplementary Table 2). Overall, HAPI analysis identified known cancer-related genes that are highly interactive with enhancers in a cancer-type specific manner.

## HAPI analysis identifies known and novel enhancer-hijacking events

HAPI analysis not only defines genes that are highly interactive with enhancers but also determines the location of their enhancers. This enables us to determine whether the enhancers are in the same genomic region as the HAPI gene or are hijacked from other genomic regions likely through structural alterations (Fig. 2A). These include both "trans" hijacking events that relocate enhancers from other chromosomes to a HAPI gene and "cis" hijacking events that relocate distant enhancers (>2 Mb, the upper limit of typical TAD domains[37]) on the same chromosome.

We analyzed H3K27ac HiChIP data of the 34 cancer cell lines to determine the origin of enhancers for HAPI genes. HAPI genes with over 25% of enhancer activity coming from either trans- or abnormal cis-interactions were treated as candidate genes driven by enhancer hijacking. In total, we identified enhancer-hijacking genes in 29 of the 34 cell lines (Fig. 2B). In this study, we focus on trans-enhancer hijacking events because they are more likely caused by structural alterations than cis-enhancer hijacking events that may be caused by other types alterations such as epigenetic loss of TAD boundaries[38,39]. We found trans-enhancer hijacking events in 16 of the cancer cell lines, but in neither of the two normal cell lines we examined (Fig. 2C). The calculated enhancer origins, both trans and cis, for all HAPI genes are listed in Supplementary Data 1.

In addition to enhancer origins, we also examined copy numbers of the HAPI genes using published SNP-array data[40]. We first focused on HAPI genes that exhibit neutral copies or modest amplifications ($\log_2(CN) < 2$) (Fig. 2D, E). This analysis identified previously reported trans-enhancer hijacking events activating oncogenes *CCND1* in the B-cell lymphoma cell line REC1[10,12] and *ETV1* in the prostate cancer cell lines LNCaP and MDAPCA2B[20,41,42]. We also found novel trans-enhancer hijacking events activating known oncogenes including *MYC* in the small cell lung cancer cell line NCIH146, *NOTCH1* in the T-cell lymphoblastic leukemia cell line CUTLL1, and *CCND1* in the breast cancer cell line ZR751. Furthermore, HAPI analysis identified potentially novel oncogenes that hijack enhancers in trans, such as *CRKL* in the colorectal cancer cell line HT29 and *FOXJ2* in the small cell lung cancer cell line NCIH2171 (Figs. 2D and S2A). Finally, we found *cis*-enhancer hijacking events activating potentially novel oncogenes such as *ID4* in the prostate cancer cell line VCaP and *TXLNA* in the small cell lung cancer cell

line CORL88 (Fig. 2E), which are likely caused by large deletions based on the known patterns of structural variants in the HiChIP signal (Figure S2B–D). Overall, cell lines with these enhancer-hijacking events are associated with higher expression of the involved genes as compared to other cell lines of the same lineage (Figs. 2D, E and S2A).

## Functional validation of identified enhancer hijacking events

We then chose two representative enhancer-hijacking genes identified from HAPI analysis for functional validation. We identified *ETV1* (chromosome 7) as a trans-enhancer hijacking gene in both LNCaP and MDAPCA2B cells, with most of its trans-enhancer activity coming from chromosome 14 (Fig. 3A). This is caused by a cryptic insertion in LNCaP and a balanced translocation in MDAPCA2B[43] that connect the two chromosomal regions. A translocation linking these two regions is also identified in a primary prostate cancer patient tumor sample from Pan-Cancer Analysis of Whole Genomes (PCAWG) (Fig. S3A). Previous studies have identified various potential enhancers within regions of this translocation, but a consensus regarding which one regulates *ETV1* has not yet been reached[20,42]. Based on H3K27ac HiChIP and ChIP-seq signal, we observed four enhancers e1-e4, upstream of *FOXA1* on chromosome 14, hijacked to the *ETV1* promoter in LNCaP and MDAP-CA2B cells (Fig. 3A). CRISPRi assays in LNCaP showed that repression of e3 and e4 caused the most significant decreases in expression of *ETV1* (Fig. 3B). H3K27ac ChIP-seq performed after CRISPRi confirmed the on-target effects (Fig. S3B). As a control, repression of the four enhancers did not affect *ETV1* expression in 22RV1 cells that lack the translocation (Fig. S3C). Interestingly, these enhancers have little or modest regulatory effects on the nearby gene *FOXA1* or *TTC6* on their original chromosome in LNCaP cells (Figs. 3B and S3D), indicating that these two genes may be dependent on additional enhancers.

*ETV1* was reported as an androgen-responsive gene in LNCaP cells promoting cell invasion[44]. We observed binding of androgen receptor (AR) to both e3 and e4 but not the promoter of *ETV1* (Figs. 3A and S3E). Additionally, a 40% upregulation of *ETV1* expression was observed in response to Dihydrotestosterone (DHT) treatment in LNCaP cells based on published RNA-seq data[45]. In contrast, *ETV1* expression is minimally affected by DHT in the untranslocated 22RV1 cells, where AR binds the e3 and e4 enhancers in their original locus (Fig. S3F). These data suggest that the androgen-induced *ETV1* upregulation is caused by these two hijacked AR-bound enhancers.

The second representative enhancer hijacking gene we selected from HAPI analysis for functional validation was *CCND1*. *CCND1* (chromosome 11) is known to hijack enhancers from chromosome 14 in the B-cell lymphoma cell line REC1[10,12]. We found that the same oncogene hijacks different enhancers from chromosome 8 in the breast cancer cell line ZR751 (Fig. 3C). WGS analysis revealed a translocation that places these enhancers upstream of *CCND1* in ZR751 (Fig. 3C). In primary breast cancer patient tumor samples from PCAWG, although we did not observe the same translocation event, we found that the *CCND1* locus is highly subject to translocations with diverse regions of chromosome 8 (Figure S3G). We applied CRISPRi to repress five of the strongest enhancers (e1–e5 in Fig. 3C) individually or simultaneously in ZR751. While targeting individual enhancers yields little (e1 and e2) or modest effect (e3, e4, and e5) on *CCND1* expression, targeting the combination of all enhancers with CRISPRi caused a 45% reduction in *CCND1* expression (Fig. 3D). Repressing most of these enhancers caused significant decreases in expression of their endogenous target gene *DUSP4* on chromosome 8 (Fig. 3D). These results together demonstrate that these enhancers activate *CCND1*, in a combinatorial manner, due to enhancer hijacking while also acting in their canonical role of activating *DUSP4*. *CCND1* is overexpressed in 50% of breast cancers, which has been partially attributed to gene amplification[46]. It is possible that enhancer hijacking may serve as an additional mechanism for *CCND1* overexpression in breast cancer.

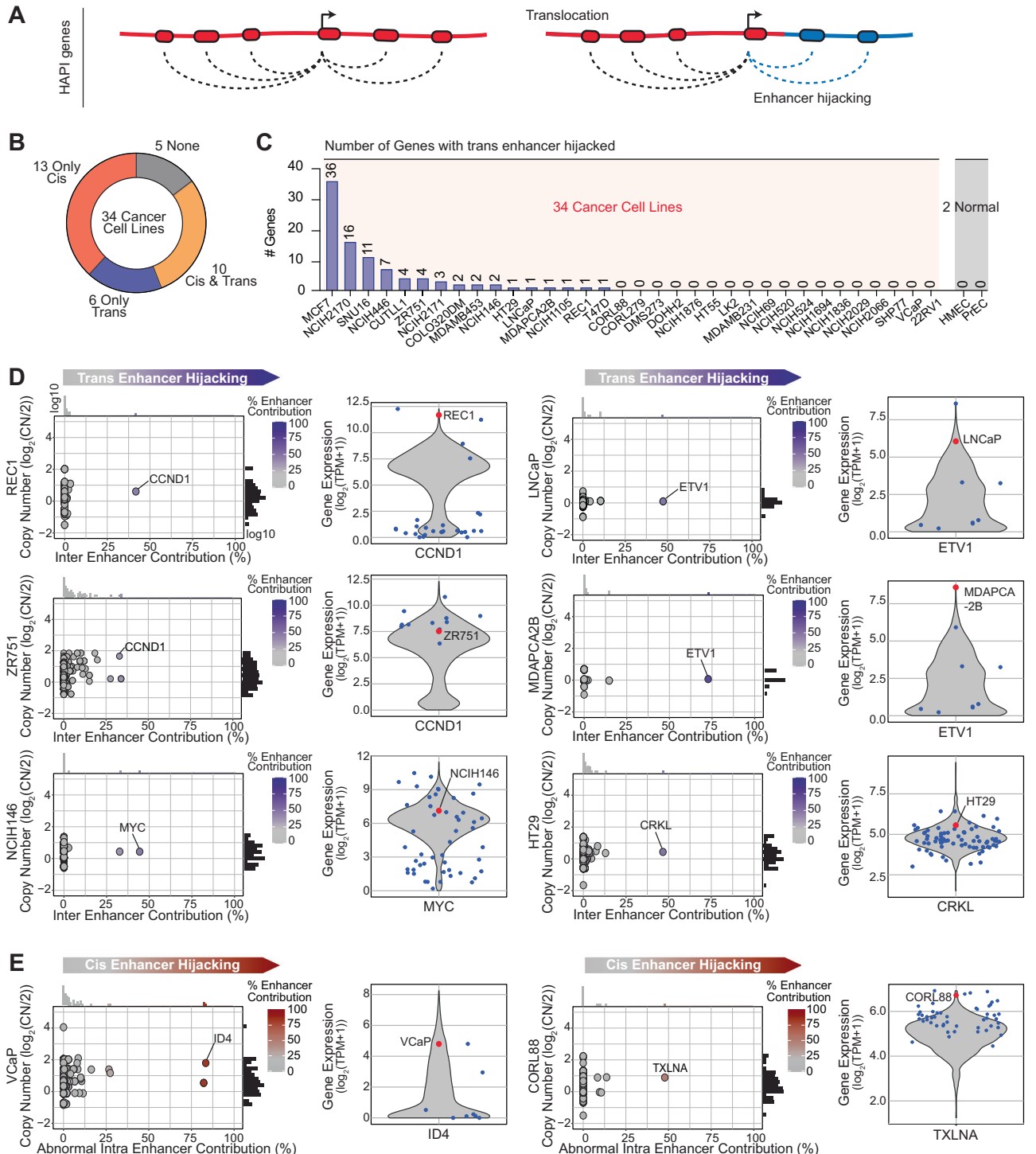

**Fig. 2 | HAPI analysis identifies known and novel enhancer-hijacking events.**
**A** Schematic models illustrating the identification of enhancer hijacking HAPI genes based on the enhancer origin. **B** The number of cancer cell lines that are identified to contain trans- and cis-enhancer hijacking events in our cohort. **C** The number of identified trans-enhancer hijacking genes found in each cancer cell line in our cohort. Two immortalized epithelial cell lines are used as negative controls.

**D** Plotted are the calculated trans-enhancer contribution (x-axis) and the copy number estimated by SNP-array data (y-axis) for each HAPI gene in selected cell lines. Besides are the expression levels for the highlighted gene in all CCLE cell lines (violin plot), the trans-enhancer hijacking cell line (red), and other cell lines of the same lineage (blue). **E** Same as D but with regards to two cis-enhancer hijacking genes in VCaP and CORL88 cells.

The hijacked enhancers on chromosome 8 in ZR751 exhibit H3K27ac enrichment in the breast cancer cell line T47D without the translocation (Figure S4A), but little enrichment in the B-cell lymphoma line REC1 in which *CCND1* hijacks enhancers from chromosome 14 (Fig. 3C). The reciprocal pattern was observed for the enhancers hijacked in REC1 (Fig. S4A). These results indicate that the enhancers involved in hijacking are already active in cancer cells without the translocation and are cancer-type specific. To support this finding, we utilized TCGA ATAC-seq data to estimate potential regulatory activity of translocated enhancers we identified in several breast, prostate, and colon cancer cell lines. Overall, these enhancers have increased ATAC-seq signal in tumors of their respective cancer types compared to

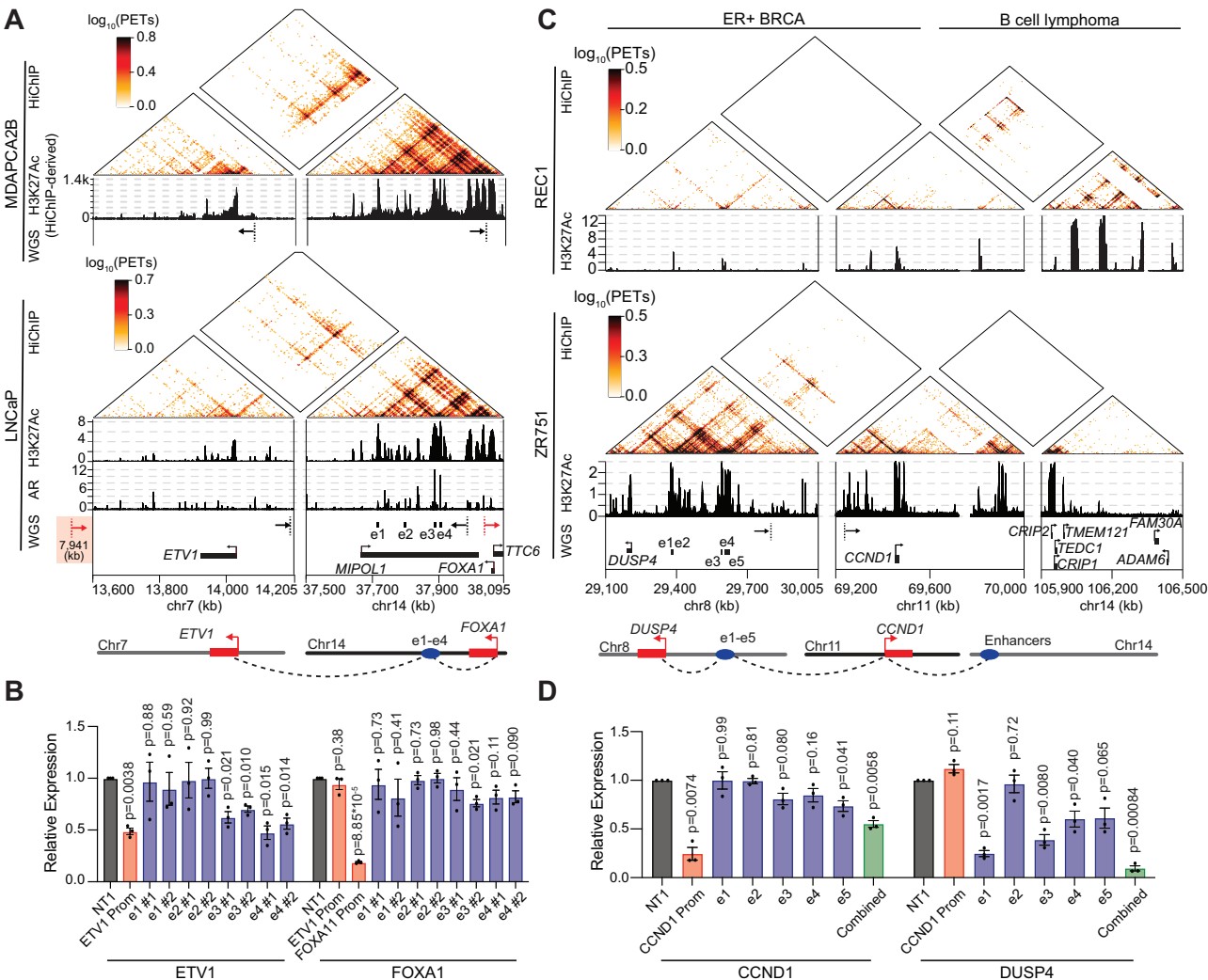

**Fig. 3 | Functional validation of identified trans-enhancer hijacking events.**
**A** HiChIP and ChIP-seq signal at chromosomal regions containing *ETV1* and the hijacked enhancers in MDAPCA2B and LNCaP cells. The H3K27ac ChIP-seq signal in MDAPCA2B cells is derived from HiChIP reads. In LNCaP cells, WGS-identified breakpoints support translocations underlying the enhancer-hijacking events. The DNA position of the left breakpoint on chromosome 7 in LNCaP cells is noted. **B** RT-qPCR measuring expression changes of *ETV1* and *FOXA1* after CRISPRi of each enhancer e1–e4 or the promoter of *ETV1* in LNCaP cells. Expression levels were normalized to cells treated with a non-targeting sgRNA (NT1). *N* = 3 biological replicates. Data are presented as mean values ± SEM. *P* values were derived from

two-sided t-tests. Source data are provided as a Source Data file. **C** HiChIP and ChIP-seq signal at chromosomal regions containing *CCND1* and the hijacked enhancers in REC1 and ZR751 cells. In ZR751 cells, WGS-identified breakpoints support translocations underlying the enhancer-hijacking events. **D** RT-qPCR measuring expression changes of *CCND1* and *DUSP4* after CRISPRi of each enhancer e1–e5, the promoter of *CCND1*, or a combination of the five enhancers in ZR751 cells. Expression levels were normalized to cells treated with a non-targeting sgRNA (NT1). *N* = 3 biological replicates. Data are presented as mean values ± SEM. *P* values were derived from two-sided t-tests. Source data are provided as a Source Data file.

others (Fig. S4B). These data together suggest that cancer-type specific enhancers may be subject to translocations to activate oncogene expression.

## HAPI analysis identifies enhancer hijacking associated with known ecDNAs

Some of the identified enhancer hijacking genes exhibit high copies (log2(CN) > 2). Recent studies have revealed that oncogenes amplified in the form of ecDNAs tend to exhibit higher copies compared to their chromosomal amplifications[47]. As illustrated in Fig. 4A, the high copy number of ecDNAs, along with increased chromatin accessibility, results in the overexpression of oncogenes such as *EGFR* in glioblastoma[47]. In addition, enhancer hijacking, as a result of complex rearrangements on ecDNAs, also contributes to oncogene expression as exemplified by *MYCN* in neuroblastoma[15]. Among our analyzed cell lines, the colorectal cancer cell line COLO320DM and the gastric

adenocarcinoma cell line SNU16 are known to contain ecDNAs amplifying the *MYC* oncogene[48]. HAPI analysis identified enhancer hijacking events in both the cell lines, which involve multiple oncogenes on the same ecDNA.

COLO320DM contains an ecDNA harboring *MYC*, a *PVT1-MYC* fusion, and several other genes including *CDX2* and *PDX1*[48]. As a HAPI gene, *MYC* has most of its enhancer activity contributed in *cis*. Interestingly, *CDX2*, a colorectal cancer lineage-survival oncogene[49,50] from chromosome 13, is identified as a trans-enhancer hijacking HAPI gene (Fig. 4B), with over 95% of enhancer activity in *trans*. Indeed, most of the enhancer activity for *CDX2* comes from the *MYC* locus on the ecDNA (Fig. 4C–E). These results suggest two separate mechanisms through which the ecDNA causes overexpression of oncogenes in COLO320DM cells—while *MYC* uses its co-amplified endogenous enhancers, *CDX2* hijacks *MYC* enhancers on the same ecDNA to enhance its expression, representing an "opportunistic" enhancer hijacking event (Fig. 4D).

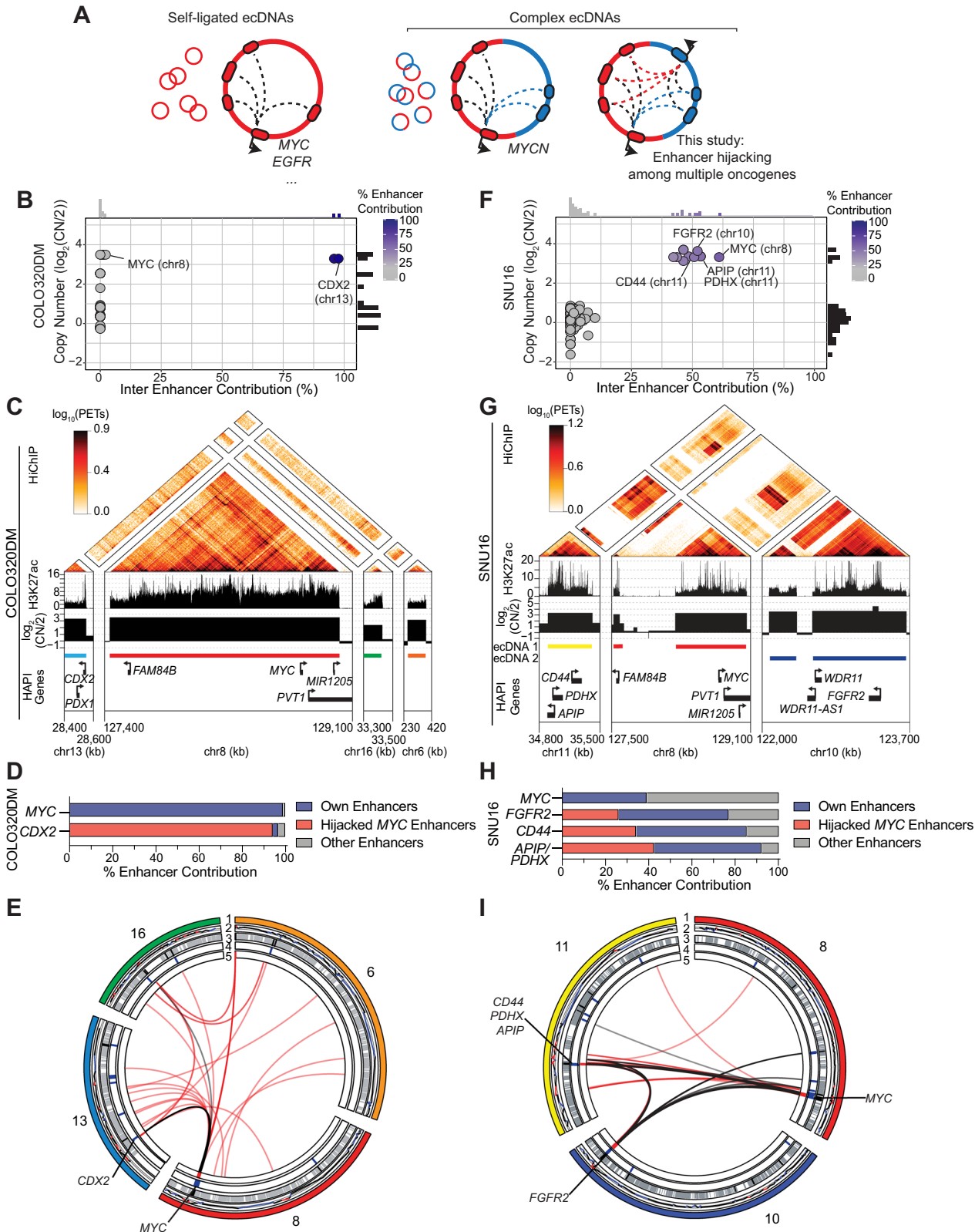

SNU16 contains multiple ecDNA isoforms harboring the *MYC* and *FGFR2* oncogenes separately or together[48,51]. The *MYC* ecDNA harbors three other genes from chromosome 11, including *APIP*, *PDHX*, and *CD44*, which are involved in the development of gastric adenocarcinoma[52,53]. HAPI analysis identified these three genes as enhancer hijacking genes (Fig. 4F), which have 34%-42% of enhancer activity contributed from enhancers near *MYC* on chromosome 8 that

are now on the same ecDNA (Fig. 4G–I). Vice versa, *MYC* also has most of the enhancer activity contributed in trans from other chromosomal regions on the same ecDNA (Fig. 4G–I). These reciprocal enhancer-oncogene interactions represent a "mutualistic" enhancer hijacking event on ecDNAs (Fig. 4H). Interestingly, although most of the *FGFR2* ecDNA does not include regions from other chromosomes based on previous reports[48], it has around half of enhancer activity contributed

**Fig. 4 | HAPI analysis identifies enhancer hijacking on known ecDNAs.**
**A** Schematic models illustrating the identification of enhancer hijacking events on ecDNAs that involve multiple oncogenes from different chromosomal regions. **B** Plotted are the calculated trans-enhancer contribution (x-axis) and the copy number estimated by SNP-array data (y-axis) for each HAPI gene in COLO320DM cells. **C** HiChIP, ChIP-seq, and SNP-array-based copy number signal at the DNA segments of the ecDNA found in COLO320DM cells. The HiChIP data and the ecDNA segments in COLO320DM were published by ref. 48. **D** Enhancer distribution of *MYC* and the additional oncogenes harbored in the ecDNA found in COLO320DM cells. Enhancers within 2 mb of distance to the promoter of *MYC* or

any other gene in the endogenous chromosomal locus will be considered as the gene's "own" enhancers, otherwise will be considered as "other" enhancers. Source data are provided as a Source Data file. **E** Circos plot: layer 1=chromosome, 2 = CNV, 3=enhancers (gray) and super-enhancers (black), 4 = HAPI TSS, 5=ecDNA segments, center=E-P loops (black) and WGS-predicted translocations (red) in COLO320DM cells. **F–I** Same as (**B–E**) but with regards to the ecDNAs in SNU16 cells. The HiChIP data and the ecDNA segments in SNU16 were published by ref. 48. Note that the *APIP* and *PDHX* genes share the same promoter anchor for the HiChIP analysis. Source data for the (**H**) are provided as a Source Data file.

---

in trans (Fig. 4F, G). This is likely caused by the reported intermolecular interactions between *MYC* and *FGFR2* ecDNAs as well as the presence of chimeric *MYC-FGFR2* ecDNA in a small proportion of the ecDNAs in this cell line[48].

## Enhancer hijacking among multiple oncogenes is prevalent on ecDNAs and complex chromosomal amplicons

We reasoned that enhancer hijacking, together with high copy numbers of the involved genes, may indicate a unique signature for chimeric ecDNAs or complex chromosomal amplicons that contain multiple DNA segments in our HAPI analysis results. Indeed, we found three additional cell lines NCIH2170, NCIH446, and MCF7 that have clusters of highly amplified HAPI genes hijacking enhancers from different chromosomes (Fig. 5A–C). We analyzed their WGS data using AmpliconArchitect, an algorithm to reconstruct amplicons[54], which showed extensive rearrangements among the amplified regions harboring the enhancer-hijacking genes (Fig. S5A–C).

In the lung squamous cell carcinoma cell line NCIH2170, we identified a cluster of trans enhancer hijacking genes from chromosomes 8 and 17 (Fig. 5A). AmpliconArchitect analysis of WGS data predicted an ecDNA containing most of these HAPI genes and their hijacked enhancers (Fig. S6A, B). In particular, we found that *ERBB2* (chromosome 17) and its surrounding oncogenes *MIEN1* and *IKZF3* strongly interact with *MYC*'s enhancers (Fig. 5D, E). *ERBB2* was known to be an amplified and overexpressed gene in this cell line[55,56]. Our analysis showed that, in addition to the copy number amplification, hijacking *MYC* enhancers on the same ecDNA may also contribute to *ERBB2* and its surrounding oncogenes' overexpression.

Additionally, in the small-cell lung cancer cell line NCIH446, HAPI analysis identified several genes including known oncogenes *MYC* and *NFIB* with trans enhancer hijacking from regions of chromosomes 8 and 9 (Fig. 5F, G). WGS analysis showed translocations among these gene loci (Fig. S6C). A recent study reported that the *MYC* locus in NCIH446 resides on the chromosome based on DNA Fluorescent In Situ Hybridization (FISH) results[57], which together with our results suggest that complex chromosomal amplicons are also capable of causing multiple oncogenes to hijack each other's enhancers.

Lastly, we identified 36 HAPI genes with trans enhancer hijacking in the ERα-positive breast cancer cell line MCF7, most of which are clustered within genomic regions of chromosomes 17 and 20 (Fig. S7A). These include known oncogenes *TBX2* and *BRIP1* from chromosome 17, and *ZNF217* from chromosome 20. Accordingly, ampliconArchitect analysis predicted an ecDNA harboring most of these HAPI genes and their hijacked enhancers (Figs. S7B, C). These oncogenes are highly interactive with each other's enhancers, indicating extensive enhancer hijacking spanning the ecDNA segments (Figs. S7D).

In the aforementioned five cell lines that harbor ecDNAs or chromosomal complex amplicons, we observed both "opportunistic" (COLO320DM and NCIH2170) and "mutualistic" (MCF7, SNU16 and NCIH446) enhancer hijacking phenomena among multiple oncogenes on the same amplicon, which are associated with overexpression of the associated oncogenes (Fig. S8A). Using publicly available AmpliconArchitect results from WGS data of primary patient tumors included

in the PCAWG project[58], we identified 1299 amplicons containing previously annotated oncogenes. We found that most of these amplicons, either in the extrachromosomal or chromosomal forms, harbor additional DNA segments originating from other chromosomal regions. In 37.6% of these complex amplicons, we found multiple oncogenes from separate DNA segments that are now on the same amplicon (Fig. 5H). One amplicon is predicted to be a *MYC-ERBB2* chimeric ecDNA, similar to what we report in NCIH2170 cells (Fig. S8B). These analyses suggest that many of these amplicons could enable extensive enhancer hijacking among their harbored oncogenes in patient tumors.

## Functional exploration of chimeric ecDNA-associated enhancer hijacking

To validate the presence of chimeric ecDNAs that contain multiple oncogenes from different chromosomes, we performed DNA FISH in NCIH2170 cells using two probes targeting the *MYC* and *ERBB2* regions separately. DNA FISH on metaphase spreads showed the presence of ecDNAs containing both *MYC* and *ERBB2* across examined cells (Fig. 6A). However, some of the ecDNAs contain only *MYC* or *ERBB2*, suggesting the intracellular heterogeneity of ecDNA compositions similar to what was reported previously in SNU16[48]. DNA FISH in interphase nuclei showed similar overlap between *MYC* and *ERBB2* signal across the examined cells (Fig. S9A). We also observed the presence of ecDNA aggregates in some of the interphase nuclei, indicative of ecDNA hubs as previously reported in other cancer cell lines[48] (Fig. S9B). These results demonstrate the presence of highly abundant *MYC-ERBB2* chimeric ecDNA in NCIH2170 cells.

The presence of chimeric *MYC* ecDNAs may provide novel therapeutic opportunities. For instance, although MYC as a transcription factor is difficult to target by small molecules, ERBB2 is considered highly targetable. Indeed, NCIH2170 exhibits the strongest sensitivity to HER2 inhibitors lapatinib and neratinib among all the lung squamous cancer cell lines tested in the Cancer Therapeutics Response Portal CTD[2] project (Fig. 6B).

We then sought to assess the regulatory function of enhancers on the *MYC-ERBB2* ecDNA. Due to the large number of potential *MYC* enhancers that are shared by these genes (Fig. 5D), individually repressing these enhancers may not reflect their combinatorial activity. As an alternative strategy, we used CRISPRi to repress the *MYC* promoter and released all *MYC* enhancers from *MYC* to assess their effect on expression of *ERBB2* and its neighboring genes that also use these enhancers (illustrated in Fig. 6C). HiChIP assays showed that CRISPRi resulted in an overall decrease in interactions between the *MYC* promoter and its enhancers, as expected (Fig. 6D–F). In contrast, *ERBB2* gained interactions with these enhancers after CRISPRi of the *MYC* promoter (Fig. 6E, F). RNA-seq showed that CRISPRi represses *MYC* expression and causes a significant increase in expression of *ERBB2* (2.7 folds) and its neighboring genes *MIEN1*, *PGAP3*, and *GRB7* (Fig. 6G). For the remaining affected genes, we found enrichments of MYC up- and down-regulated gene sets, suggesting that they are secondary effects of *MYC* repression (Fig. S9C). All these results demonstrate that, on the chimeric *MYC-ERBB2* ecDNA in NCIH2170 cells, while *MYC* retains its interaction with its endogenous enhancers, *ERBB2* and

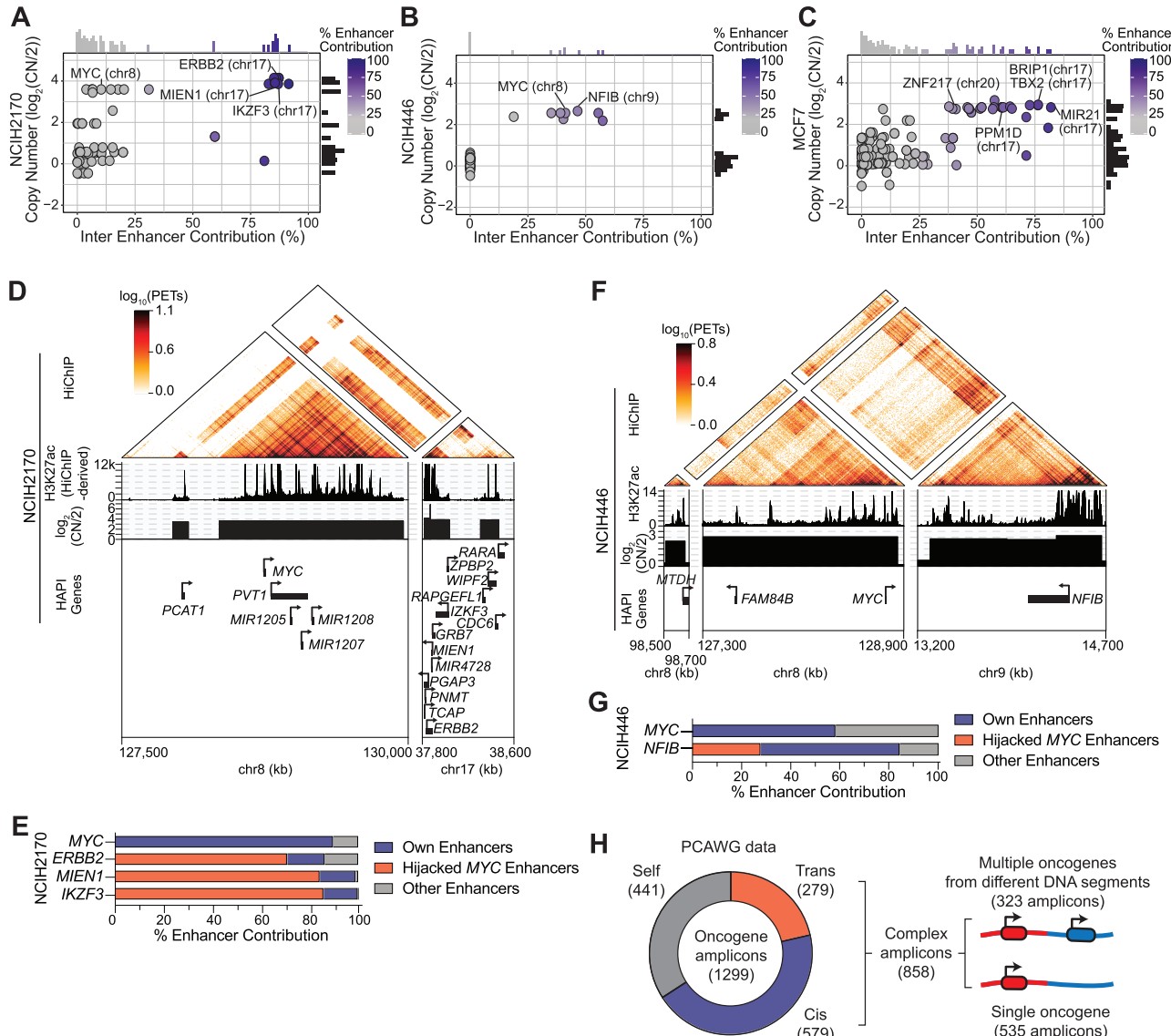

**Fig. 5 | Enhancer hijacking is prevalent on complex amplicons. A–C** Plotted are the calculated trans-enhancer contribution (x-axis) and the copy number estimated by SNP-array data (y-axis) for each HAPI gene in NCIH2170, NCIH446, and MCF7 cells. **D** HiChIP, H3K27ac ChIP-seq (derived from HiChIP reads), and copy number profiles at DNA segments of the *MYC* ecDNA in NCIH2170 cells. **E** Enhancer distribution of *MYC* and the additional oncogenes harbored in the ecDNA found in NCIH2170 cells. Source data are provided as a Source Data file. **F** HiChIP, H3K27ac ChIP-seq, and copy number profiles at DNA segments of the *MYC* amplicon in NCIH446 cells. **G.** Enhancer distribution of *MYC* and the additional oncogenes harbored in the amplicon found in NCIH446 cells. Source data are provided as a Source Data file. **H** Summary of AmpliconArchitect data from PCAWG tumor samples: most of oncogene amplicons contain additional trans or cis DNA segments. 37.6% of these complex amplicons contain multiple oncogenes originating from different DNA segments.

its neighboring genes opportunistically hijack *MYC* enhancers on the same ecDNA for their transcriptional activation.

## Discussion

Enhancer hijacking caused by structural alterations is an oncogenic event in cancer; however, it remains difficult to detect due to the complexity of cancer genomes. Here, we presented a two-step strategy, namely HAPI, to robustly detect enhancer hijacking events based on: 1) oncogenes subject to enhancer hijacking should be highly regulated by enhancers, 2) the hijacked enhancers should contribute to a substantial proportion of an oncogene's enhancer activity. This assumption is supported by recent work using CRISPR to generate de novo enhancer translocations to the *MYC* locus, which showed that the combination of enhancer activity and contact from the translocated regions determines *MYC* activation[59]. Although our analysis was based

on H3K27ac HiChIP data, we expect it to be compatible with other types of high-resolution enhancer-promoter contact mapping assays such as PLAC-seq, Micro-C, and ChIA-PET[60–62].

We first found hundreds of HAPI genes in each cell line whose promoters are highly interactive with enhancers. Previous work focused on neural cells found that genes that specify lineage development exhibit this pattern[63]. Our analysis in cancer cell lines found that oncogenes are also enriched in this category. We then prioritized HAPI genes that have a substantial amount of enhancer activity originating from other chromosomes or abnormally distant intra-chromosomal regions. This led to the identification of known and potentially novel enhancer-hijacking events regardless of the underlying structural variants such as balanced translocation, cryptic insertion, large deletion, as well as ecDNA formation. Although several of the identified oncogenes such as *MYC*, *ETV1*, and *CCND1* have been previously linked to enhancer

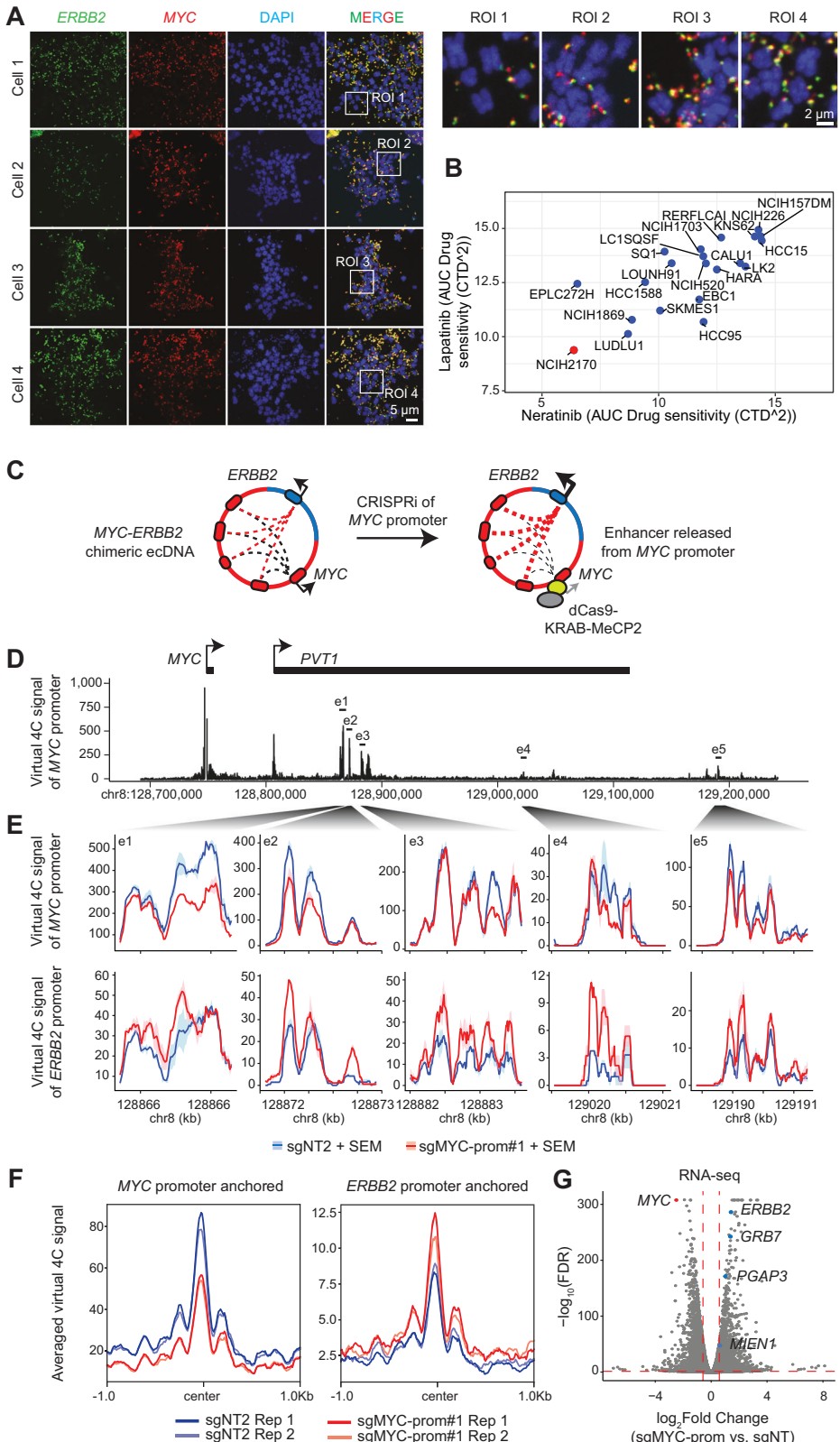

hijacking, we identified their novel hijacked enhancers that are cancer type specific. The analysis also identified potentially novel enhancer-hijacking oncogenes such as *CRKL* and *ID4*. We have made the HAPI analysis tool publicly available (https://doi.org/10.24433/CO.1110712.v1) and anticipate its broader application in cancer driver discovery and mechanism investigation.

In addition to individual enhancer-hijacking events caused by simple DNA rearrangements, we identified nested enhancer-hijacking on complex amplicons, such as ecDNAs, that involve multiple oncogenes originating from different chromosomal regions. Our analysis of previously characterized as well as our newly identified ecDNAs shows that ecDNAs may contain multiple oncogenes from different

**Fig. 6 | Functional exploration of the *MYC-ERBB2* chimeric ecDNA.**
**A** Representative images of metaphase chromosome spreads showing highly amplified copies of *MYC* (red) and *ERBB2* (green) colocalizing outside of chromosomal DNA (blue, DAPI staining) in NCIH2170 cells. In total, 20 metaphase spreads of high quality were collected from seven separate images, all indicating the presence of *MYC-ERBB2* chimeric ecDNAs in high abundance. Scale bar, 5 μm. Four enlarged regions of interest (ROI) were highlighted. Scale bar, 2 μm. **B** AUC values representing the sensitivity of lung squamous carcinoma cell lines to Neratinib and Lapatinib. Source data are provided as a Source Data file. **C** Schematic illustration of the experimental design of the *MYC*-promoter CRISPRi assay. **D** Virtual 4 C signal (derived from HiChIP data) showing interactions with the *MYC* promoter at the chromosome 8 locus (now on ecDNA) in NCIH2170 cells (sgNT2, replicate 1).

**E** Virtual 4 C signal between the promoter of *MYC* (upper) or *ERBB2* (lower) and five highlighted enhancer regions representing different levels of interaction intensity in NCIH2170 cells with and without *MYC* promoter repression (sg*MYC*-prom vs. sgNT2). SEM: standard error of the mean. *N* = 2 biological replicates. **F** The averaged virtual 4 C signal between the top 50 *MYC*-interacting regions from the *MYC* locus (ranked by their interactions with the *MYC* promoter) and the promoter of *MYC* (left) or *ERBB2* (right). **G.** A volcano plot presenting RNA-seq results from NCIH2170 cells with and without *MYC* promoter repression. Two separate sgRNAs were used for each condition (sgNT1/sgNT2, sg*MYC*-prom#1/ sg*MYC*-prom#2). Highlighted are the genes on the *MYC-ERBB2* ecDNA that are significantly differentially expressed.

chromosomes, which we refer to as chimeric ecDNAs. These structures were predicted by WGS analysis and validated by either optical imaging from previous work[48,64] or dual-color DNA FISH in this study. We revealed two types of enhancer hijacking events associated with chimeric ecDNAs, as exemplified by *MYC* ecDNAs in this study. For the *MYC* ecDNAs found in COLO320DM and NCIH2170 cells, while *MYC* mainly uses its own enhancers, the additional oncogenes such as *CDX2* and *ERBB2* that are translocated to the *MYC* ecDNAs heavily use *MYC* enhancers for their activation, representing an "opportunistic" enhancer hijacking event. For the *MYC* ecDNA found in SNU16 cells, *MYC* and the co-amplified *CD44* gene use each other's enhancers in a more balanced manner, representing a potentially "mutualistic" enhancer hijacking event. These results provide important insights about another layer of complexity for oncogene amplification and overexpression.

Chimeric ecDNAs may be prevalent in cancer. A recent work reported that 33.1% of ecDNAs found in Barret's esophagus and esophageal adenocarcinomas contain multiple oncogenes on the same molecule based on WGS analysis[65]. In addition to ecDNAs, chromosomal translocations often precede amplifications, also causing co-amplification two oncogenes from distinct chromosomal regions[66]. Characterization of these chromosomal or extrachromosomal complex amplicons may reveal novel therapeutic strategies. As exemplified in this study, although *MYC* is a notoriously difficult drug target, *ERBB2* that is co-amplified with MYC is highly targetable.

Our observation of potentially "mutualistic" or "opportunistic" enhancer hijacking events on different chimeric ecDNAs has important implications for the future of CRIPSR-based therapeutics being developed to silence oncogenes. Targeting the shared enhancers on chimeric ecDNAs that contribute to overexpression of multiple oncogenes is an appealing therapeutic strategy to repress multiple oncogenes simultaneously. However, we also demonstrate that repressing the *MYC* promoter on a chimeric ecDNA released the shared enhancers from *MYC* and increased *ERBB2's* opportunistic enhancer hijacking and expression. This could be a strategy to further sensitize the tumor to HER2 inhibitors, or it could disastrously increase growth factor driven proliferation. These examples illustrate the importance of identifying and characterizing these complex regulatory events involving multiple oncogenes in order to develop effective CRISPR-based therapeutic strategies.

One limitation of our work is that we primarily used cell lines as our experimental models because they are relatively homogenous and easy to engineer. We used WGS data from primary human tumors in PCAWG to support our conclusions about the presence of complex rearrangements involving multiple oncogenes but the functional validation of enhancer hijacking in primary human tumors requires further development of 3D genomics assays and analysis methods that account for their intrinsic heterogeneity. In addition, we focused on trans-enhancer hijacking events that are caused by structural alterations. Many cis-enhancer hijacking events may be caused by other types of genomic alterations such as point mutations and small indels as well as epigenetic alterations such as DNA methylation that are capable of disrupting TAD domains, which fall outside the scope of this manuscript. Future

efforts integrating additional model systems and genomic/epigenomic analysis may address these limitations.

In summary, applying a two-step HAPI analysis, we identified and characterized enhancer-hijacking events that are caused by chromosomal and extrachromosomal structural alterations in cancer. Our findings reveal enhancer-hijacking events underlying the activation of multiple oncogenes that reside together on complex amplicons such as ecDNAs.

## Methods
### Cell lines
Prostate cancer cell lines LNCaP, MDAPCA2B, 22Rv1, and VCaP, ERα-positive breast cancer cell lines ZR751 and MCF7, lymphoma cell lines REC1 and DOHH2, lung squamous cell line NCIH2170, and small cell lung cancer cell lines NCIH446 and NCIH1876 were obtained from the Cancer Cell Line Encyclopedia (CCLE)[67,68]. The lymphoma cell line CUTLL1 was obtained from Sigma. VCaP, MCF7, and MDAPCA2B cells were grown in DMEM media supplemented with 10% FBS. NCIH1876 cells were grown in DMEM/F12 media supplemented with 4 μg/ml Hydrocortisone, 5 ng/ml murine EGF, 1× Insulin-Transferrin-Selenium, 10 nM of beta-estradiol, and 10% FBS. The rest of the cell lines were grown in RPMI media supplemented with 10% FBS. The identities of the cell lines were verified by either SNP fingerprinting as previously described in the CCLE project[67,68] or short tandem repeat analysis through IDEXX. All the cell lines were tested negative for mycoplasma.

### HiChIP and loop calling
HiChIP assays were performed for the following cell lines: CUTLL1, REC1, VCaP, MDAPCA2B, 22Rv1, ZR751, MCF7, DOHH2, NCIH446, NCIH1876, and NCIH2170 as described in the published protocol[69] with minor modifications described previously[70]. Cross-linked chromatin was digested with the MboI restriction enzyme, filled with dNTPs including biotin labeled dATP, ligated with T4 DNA ligase, and sonicated to obtain chromatin fragments averaging ~1 kb in length using Qsonica (Q800). To enrich for chromatin interactions at active regulatory elements, antibodies of H3K27ac (Abcam, ab4729, rabbit polyclonal, 7.5 μg/HiChIP) were used for DNA fragment capture. Magnetic streptavidin beads were applied to pull down fully ligated DNA fragments and HiChIP libraries were prepared using Illumina Tagment DNA Enzyme and Buffer Kit. Sequencing was done with Illumina NextSeq or NovaSeq. HiChIP data from the other cell lines included in the study was obtained from the NCBI Gene Expression Omnibus (GEO) (Supplementary Table 1).

The paired-end HiChIP sequencing reads were aligned to human genome hg19 with the HiC-Pro pipeline[71]. Hichipper was then used to call chromatin loops where DNA anchors were assigned, and the number of PETs connecting the anchors[72]. Loop anchors were either called by hichipper based on enrichment of HiChIP sequencing reads or derived from broadpeaks called by MACS2 using publicly available H3K27ac ChIP-seq data in the corresponding cell lines, as indicated in Supplementary Table 1. If multiple biological replicates of HiChIP data are available, the reads of replicates were merged prior to analysis.

Chromatin loops were removed if their PETs <3 or anchors overlap with ENCODE blacklist regions[73]. HiChIP data were visualized using the R package gTrack[3] (URL: https://github.com/mskilab/gTrack) or RCircos[74]. The gene positions in gTrack plots are based on the GENCODE database.

## HAPI analysis

We grouped the loops, called by hichipper, into "normal," "abnormal cis", and "trans" groups. Intra-chromosomal chromatin interactions spanning more than 5 kb and less than 2 mb were considered normal loops. Intra-chromosomal interactions, whose anchors are separated by at least 2 mb were considered abnormal long-range cis chromatin loops. Inter-chromosomal interactions were treated as trans chromatin loops. The H3K27ac HiChIP anchors were then intersected with ±2.5 kb of transcription start sites (NCBI RefSeq gene list downloaded from UCSC browser in 2019) using bedtools[75] to identify promoters and enhancers. We consider a HiChIP loop that has a promoter in only one of its anchors as an enhancer-promoter loop. As promoters may act as enhancers or repressors for another gene[76–78], we excluded promoter-promoter interactions for our analysis.

To identify highly interactive genes, all types of enhancer-promoter interactions (normal, abnormal cis, and trans) described above were combined. Then for every gene, two scores were calculated to account for the number and strength of interactions between a promoter and its enhancers. The first score, defined as an enhancer contact value, is the number of enhancer anchors that are looped to the promoter of a gene. The second score, defined as interaction intensity value, is the summation of the paired-end tags (PETs) that support the interactions of the promoter and its enhancer anchors. In the case of genes that have multiple promoters, the promoter with the most PETs stemming from it is used as the primary anchor for enhancer–promoter interactions. Genes are then ranked based on these scores individually and are presented in hockey stick plots. The inflection points of these plots, defined as the point where the line with a slope of ((max value-min value)/number of values) is tangent to the curve, are the cutoff values to define genes with a significant enhancer contacts values or interaction intensity. Genes with both scores greater than or equal to their respective cutoffs are defined as HAPI genes.

To determine what types of enhancers are interacting with HAPI genes, the trans and long-range cis interaction data sets with enhancer contact values and interaction intensity calculated for each gene as described above are used. Then a percent enhancer contribution is calculated for each HAPI gene using the following formula for both trans and long-range cis interaction data:

$$\text{Enhancer Contribution} = \frac{\log 10(\text{cis OR trans PETs})*\text{cis OR trans anchors}}{(\log 10(\text{local PETs})*\text{local anchors}) + (\log 10(\text{cis PETs})*\text{cis anchors}) + (\log 10(\text{trans PETs})*\text{trans anchors})}$$

Using this formula, genes with a higher trans enhancer contribution have a higher number of interacting enhancers and stronger interaction intensity coming from a different chromosome. Similarly, genes with a higher abnormal enhancer contribution have a higher number of enhancers and stronger interaction intensity coming from regions greater than 2 mb away on the same chromosome. We focus on the genes that have more than 25% of enhancer activity contributed in trans or cis.

To calculate the contributions of enhancers hijacked from the *MYC* locus to a target gene (Figs. 4D, 4H, 5E, and 5G), we used the same formula but classified the enhancers into "own enhancers" (cis-interacting enhancers within 2 mb of the target gene's promoter), "hijacked MYC enhancers" (trans-interacting enhancers hijacked from regions within 2 mb of the MYC promoter), and "other enhancers".

## Copy number analysis

SNP-array-based copy number segment data of cancer cell lines were downloaded from the Broad Institute CCLE Portal (URL: https://data.broadinstitute.org/ccle_legacy_data/dna_copy_number/CCLE_copynumber_2013-12-03.seg.txt; downloaded in November, 2020). For the CUTLL1 cell line that is not included in the CCLE copy number dataset, we used Neoloop[20] to infer the copy number profile using HiChIP reads.

## Clustering analysis

An interaction score was calculated for all HAPI genes in each cell line by multiplying the $\log_{10}$ of the interaction intensity (PETs) by the enhancer contact value (interaction score = $\log_{10}$(interaction intensity) * (enhancer contact value)). The clustering analysis was based on Spearman pairwise sample correlation. Heatmap was generated using the heatmap.2 function in the gplots R package.

## Pathway analysis and oncogene enrichment analysis

For HAPI genes that are shared by at least half of the cell lines in each cancer type, we performed pathway analysis and oncogene enrichment analysis. For stomach adenocarcinoma, since only one cell line (SNU16) was included in our analysis, all its HAPI genes are used to represent this cancer type. GSEA analysis (MSigDB)[79,80] was performed to identify the top KEGG legacy pathways[81,82], Reactome pathways[83], and Gene Ontology molecular functions[84,85] enriched in the shared HAPI genes. As the GSEA 'compute overlaps' tool requires no more than 500 genes as the input, for prostate cancer cell lines that have 548 shared HAPI genes identified, we ranked the genes based on their interaction scores (averaged across the corresponding cell lines) and used the top 500 genes for the analysis. The shared HAPI genes and other genes in each cancer type were also compared to a list of known oncogenes merged from three previously published oncogene data sets[34–36]. Fisher exact tests were used to calculate oncogene enrichment in the HAPI genes.

## RT-qPCR

The Zymo Quick-RNA miniprep kit paired with on-column DNase I treatment was used to extract total RNA. To make cDNA for RT-qPCR, 1μg of RNA was used with the NEB LunaScript SuperMix kit. Real-time PCR was completed with NEB Luna Universal qPCR Master Mix on a Bio-Rad CFX96 qPCR instrument with technical replicates. The qPCR signal was first normalized to the internal reference gene *CTCF* and then the non-targeting sgRNA (sgNT1). RT-qPCR primers are listed in Supplementary Table 3.

## RNA-seq

CCLE RNA-seq data were downloaded from the DepMap portal (URL: https://depmap.org/portal/; downloaded in May, 2022)[67]. Additional RNA-seq data for the CUTLL1 cell line was downloaded from GSE61999 and aligned to hg19 using Bowtie2[86] and gene expression was calculated using RSEM[87]. The log2(TPM + 1) values of CUTLL1 were merged with the CCLE dataset for comparison.

For determining androgen dependence of *ETV1* expression in LNCaP and 22RV1, RNA-seq data for both cell lines grown in androgen-depleted media treated with DHT or vehicle were downloaded from GSE92574. Counts per million reads were used to determine the $\log_2$ fold change in expression between control and treated cells. We generated RNA-seq data for the CRISPRi experiment in NCIH2170 cells and

performed edgeR analysis[88] to identify differentially expressed genes (FDR < 0.05; Fold change > 1.5).

## ChIP-seq

ChIP-seq was performed in LNCaP-dCas9-KRAB-MeCP2 cells treated with enhancer-targeting or non-targeting sgRNAs, as previously described[70]. Five million cells were crosslinked with 1% formaldehyde (diluted in 1× PBS) and lysed with Lysis Buffer I (5 mM PIPES pH 8.0, 85 mM KCl, 0.5% NP40) and then Lysis Buffer II (1× PBS, 1% NP40, 0.5% sodium deoxycholate, 0.1% sodium dodecyl sulfate) supplemented with protease inhibitors. Chromatin extract was sonicated with Bioruptor (20 min; pulse: 30 s on/30 s off; high amplitude) and immunoprecipitated with H3K27ac antibody (Abcam, ab4729, rabbit polyclonal, 4 μg/ChIP) premixed with Dynabeads A and G. ChIP-seq libraries were prepared using NEBNext DNA Ultra II library prep kit and sequenced by Illumina MiSeq.

Remaining ChIP-seq data used in the study were obtained from previously published datasets (Supplementary Table 1). ChIP-seq reads and genomic input reads, if available, were aligned to the hg19 using Bowtie[86]. Samtools[89] was used to sort and index aligned reads followed by MACS2[90] to call significant peaks (q value < 0.05) and calculate the ChIP-seq signal. Super-enhancers were called using H3K27ac ChIP-seq signal using the Homer pipeline[91]. We focused on super-enhancers that contain at least one enhancer anchor of a HiChIP loop and linked them to gene promoters utilizing the pairToBed function of bedtools[75].

## CRISPR-mediated enhancer repression

To repress enhancers via CRISPR, we used a lentiviral dCas9-KRAB-MeCP2 vector that was previously generated[28]. Cells were infected with the vector and selected with blasticidin (10 μg/ml) for >5 days to stably express dCas9-KRAB-MeCP2. sgRNAs were designed to target the peaks of DNase I hypersensitive regions and the trough of H3K27ac ChIP-seq profiles for either enhancer or promoter regions of interest. The promoter- or enhancer-targeting sgRNAs and non-targeting sgRNAs with no genome recognition sites were cloned into LentiGuide-Puro (Addgene: 52963). The cells stably expressing dCas9-KRAB-MeCP2 were infected with these vectors and then selected with puromycin (2 μg/ml) for at least 3 days before extracting RNA. All sgRNA sequences used are listed in Supplementary Table 3.

## WGS data processing and structural variant calling

WGS data for LNCaP, MCF7, ZR751, NCIH446, NCIH2170, COLO320DM, and SNU16 were downloaded from published studies (Supplementary Table 1). Raw FASTQ files were aligned to hg19 using BWA-MEM[92] and the aligned reads were sorted and indexed with Samtools[89]. Structural variants and their breakpoints were called using GRIDSS[93,94]. The output VCF files were converted to bedpe files and filtered by QUAL ≥ 1000, AS > 0, and RAS > 0 for translocation visualization using the Bioconductor package StructuralVariantAnnotation[95].

## PCAWG structural variants

PCAWG consensus call sets of structural variants in tumor samples were downloaded from the ICGC Data Portal (URL: https://dcc.icgc.org/releases/PCAWG/consensus_sv; release date: Nov 25, 2019) and presented using RCircos[74].

For PCAWG complex amplicon analysis, we used the publicly available AmpliconArchitect results released from the AmpliconRepository (URL: https://ampliconrepository.org; downloaded in October 2023). We used the "aggregated results" file in the "PCAWG – full" dataset released on August 29, 2023, which listed the DNA segments included in each amplicon. To identify DNA segments harboring oncogenes, we used a list of known oncogenes merged from three previously published oncogene data sets[34–36].

## ecDNA calling

To call ecDNAs in CCLE cell lines we employed the AmpliconSuite pipeline (URL: https://github.com/jluebeck/AmpliconSuite-pipeline). Briefly, WGS data were aligned to hg19 using BWA[92] and amplified regions were called with cnvkit[96]. Seed regions for AmpliconArchitect[54] were identified using the default settings of the prepareAA script. AmpliconArchitect[54] was used to identify ecDNAs. AmpliconClassifier[65] was used to determine if the amplicons represent ecDNAs and CycleViz (URL: https://github.com/jluebeck/CycleViz) was used to visualize the reconstructed ecDNA.

## TCGA ATAC analysis

We downloaded the TCGA ATAC-seq data[97] from the UCSC Xena Browser (URL: https://xenabrowser.net/datapages/?dataset=TCGA_ATAC_peak_Log2Counts_dedup_sample&host=https%3A%2F%2Fatacseq.xenahubs.net&removeHub=https%3A%2F%2Fxena.treehouse.gi.ucsc.edu%3A443; version: 2018-06-21), which contains log2-transformed normalized ATAC-seq insertion counts for each tested sample across the identified pan-cancer ATAC peaks. To compare chromatin openness of the hijacked enhancers across different cancer types, we first took enhancers that are hijacked by HAPI genes in each selected cell line, and then summed the log2 normalized ATAC-seq counts from ATAC peaks overlapping with the hijacked enhancers to determine its overall chromatin openness for each sample included in the tested cancer types.

## Dual-color DNA FISH

Metaphase arrest of NCIH2170 cells was performed by incubation with KaryoMAX (Gibco) at 0.1 μg/mL overnight. Single cell suspension of the metaphase arrested cells was resuspended in 1xPBS followed by incubation with 75 mM KCL for 15-30 min. The cells were fixed with 3:1 methanol to glacial acetic acid overnight and then washed three times with the fixative before being dropped on a coverslip. The coverslip was dried for 5 min at 60 °C before being immersed in 2xSSC in preparation for DNA FISH.

Coverslips with fixed cells in metaphase were washed twice for 5 min with 2xSSC. The coverslips were then washed with 50% formamide/1×SSC for 4 min and dehydrated through a series of ethanol incubations (70%, 85%, and 100%) for 2 min each. Next, the coverslips were placed on slides containing a 1:1 ratio of ERBB2 to MYC FISH probes diluted in the manufacturer's hybridization buffer (Empire Genomics). The coverslips were sealed with rubber cement and denatured in a heated incubator at 82 °C for 1 min, followed by incubation in a humidified chamber at 37 °C overnight. The coverslip was washed once with 0.4×SSC/0.3% Igepal at 73 °C and twice with 2×SSC/0.1% Igepal each for 2 min. A final two washes with 2xSSC were performed before mounting the coverslip using Prolong Antifade Mountants with DAPI (Thermo Fisher).

For interphase DNA FISH, NCIH2170 cells were seeded on collagen-coated coverslips and fixed with 4% PFA in 1×PBS for 10 min, followed by quenching for 10 min with 0.1 M Glycine in 1×PBS. The cell membrane was then permeabilized for 10 min with 0.5% Triron-X 100. An additional three 5 min rinses with 1xPBS proceeded RNase A treatment at 0.1 ug/mL for 1 h at 37 °C, followed by three more rinses with 1×PBS before a 10 min incubation with 0.7% Triton X-100 in 0.1 M HCl on ice. After three rinses with PBS, the coverslips were incubated for 30 min in 2x SSC with 50% formamide. During the incubation, the hybridization buffer containing 2×SSC, 1 μl each of MYC and ERBB2 FISH probes (Empire Genomics), 50% (v/v) formamide, 2% dextran sulfate, and 0.2 μl 10 mg/ml salmon sperm DNA was prepared and 10 μl was placed on the slide. The coverslips were then placed on the hybridization buffer on the slide and sealed with rubber cement. Denaturation was performed in a heated incubator at 78 °C for 4 min followed by incubation overnight at 37 °C in a humidified chamber.

The coverslips were then washed three times with 50% formamide in 2×SSC for 5 min each at 37 °C. An additional three washes with 2×SSC at 37 °C were performed before mounting with Prolong Antifade Mountants with DAPI (Thermo Fisher).

All DNA FISH samples were imaged using a Nikon Eclipse Ti-2 microscope coupled with a CSU-W1 spinning disk confocal scanner unit (Yokogawa). Specifically, the fluorescent cell samples were excited using three laser lines: 405 nm, 488 nm, and 561 nm. The fluorescence signals were captured and detected using a Plan Apo 60x objective (Nikon, NA 1.40) and a high-speed Kinetix sCMOS camera (Photometrics), respectively. This microscope utilizes the Nikon Perfect Focus System, an interferometry-based focus lock that allows the capture of multipoint images without loss of focus. All imaging data were examined and quantified using Fiji and Cell Profiler.

### Virtual 4C analysis
For HiChIP data from NCIH2170 cells with and without CRISPRi-mediated *MYC* promoter repression, we presented and compared their virtual 4C signal anchored at *MYC* or *ERBB2* promoters. Briefly, we first identified all valid read pairs (HiC-Pro output) that have one end mapped to a 2 kb window centered at *MYC* or *ERBB2* TSS. We then used the other end of the read pairs to generate a bedgraph file at a 10 bp resolution. The bedgraph signal was then scaled based on the total number of PETs identified in each condition. As most of enhancers used by *MYC* and *ERBB2* come from the chromosome 8 locus surrounding *MYC*, we presented Virtual 4C signal at five regions from the *MYC* locus with a range of enhancer activity as well as the averaged Virtual 4C signal for the top 50 HiChIP signal-derived peaks at the *MYC* locus that are ranked based on their interactions with the *MYC* promoter.

### Reporting summary
Further information on research design is available in the Nature Portfolio Reporting Summary linked to this article.

## Data availability
The publicly available H3K27ac HiChIP data used in this study were obtained from the NCBI Gene Expression Omnibus (GEO) at GSE227242, GSE97585, GSE157381, GSE147854, GSE166232, GSE151002, GSE159985, and GSE188401 (specific cell lines are described in Supplementary Table 1). The publicly availably ChIP-seq data used in this study were downloaded from GEO at GSE73994, GSE63109, GSE69558, GSE105760, GSE105627, GSE85158, GSE57436, GSE86743, GSE73319, GSE115123, and GSE159972 for H3K27ac, and GSE174109 and GSE94013 for AR ChIP-seq (specific cell lines are described in Supplementary Table 1). TCGA copy number segment data for all applicable cell lines were downloaded from the Broad Institute Portal (URL: https://data.broadinstitute.org/ccle_legacy_data/dna_copy_number/CCLE_copynumber_2013-12-03.seg.txt). CCLE RNA-seq data were downloaded from the DepMap portal (URL: https://depmap.org/portal/). RNA-seq data for CUTLL1 was downloaded from GSE61999. RNA-seq data for LNCaP and 22RV1 cells with and without DHT treatments were downloaded from GSE92574. Publicly available WGS data for cell lines were downloaded from NCBI SRA: PRJNA361316, PRJNA523380, and PRJNA506071 (specific cell lines are described in Supplementary Table 1). PCAWG consensus call sets of structural variants in tumor samples were downloaded from the ICGC Data Portal (URL: https://dcc.icgc.org/releases/PCAWG/consensus_sv). Complex amplicon data from PCAWG were downloaded from the AmpliconRepository (https://ampliconrepository.org). Publicly available TCGA ATAC-seq data were downloaded from the UCSC Xena Browser (URL: https://xenabrowser.net/datapages/?dataset=TCGA_ATAC_peak_Log2Counts_dedup_sample&host=https%3A%2F%2Fatacseq.xenahubs.net&removeHub=https%3A%2F%2Fxena.treehouse.gi.ucsc.edu%3A443). The ChIP-seq, HiChIP, and RNA-seq data generated in this study have been deposited to GEO under the series GSE228247. The remaining data are available within the Article, Supplementary Information, Supplementary Data, or Source Data file. Source data are provided with this paper.

## Code availability
HAPI analysis is available for public use with test examples as a module in Code Ocean at https://doi.org/10.24433/CO.1110712.v1.

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

## Acknowledgements

We thank Matthew Meyerson, John Pulice, and Dean Tantin for insightful discussions. The support and resources from the Center for High Performance Computing and the HSC Imaging Core at the University of Utah are gratefully acknowledged. Funding for this work was provided by R00CA215244 and R37CA263505 from the National Cancer Institute (X.Z.), R35GM150941 from the National Institute of General Medical Sciences (Y.L.), 132596-RSG-18-197-01-DMC from the American Cancer Society (K.E.V.), Huntsman Cancer Institute's National Cancer Institute Cancer Center Support Grant (P30CA042014) and the Huntsman Cancer Foundation. The funders had no role in study design, data collection and analysis, decision to publish, or preparation of the manuscript.

## Author contributions

K.L.M., E.R.W., H.E.J., S.D.B., and X.Z. performed the computational analysis for the manuscript, and K.L.M., C.D., N.E.P., and X.Z. performed the wet-laboratory experiments. K.E.V. and X.Z. designed the study. J.G., S.D.B., Y.L., K.E.V., and X.Z. supervised the study. K.L.M., K.E.V., and X.Z. wrote the manuscript with input from other co-authors.

## Competing interests

The authors declare no competing interests.
