## [Peer Review File · Nature Communications]

3D genomic analysis reveals novel enhancer-hijacking caused by complex structural alterations that drive oncogene overexpressionEditorial Note: This manuscript has been previously reviewed at another journal that is not operating a transparent peer review scheme. This document only contains reviewer comments and rebuttal letters for versions considered at *Nature Communications*.

REVIEWERS' COMMENTS

Reviewer #1 (Remarks to the Author):

The authors have responded to some of my concerns with additional analysis and explanation, however, it is challenging to identify the changes added without any line- or page numbers or referencing in the rebuttal.

The section of essentially re-discovering ETV1 is still mostly incremental. Finding enhancer regions required for the activation is worthwhile to report, but overall, this section does not add much novelty and should be toned down.

A new analysis finding what appears to be competition between promoters on ecDNAs is interesting and novel and should be highlighted. In my view, this is the most interesting part of the revised manuscript. This is not a new mechanism IMO, but an interesting observation that supports the notion that E:P interactions can compete and is zero-sum (on ecDNA).

- There is no compelling evidence for a new mechanism (called "mutualistic"). ecDNAs have been shown multiple times to have relaxed chromatin and highly interacting structures. The authors need to remove statements claiming a novel mechanism here.

The authors highlight two MYC containing ecDNAs - one in COLO320 and one in SNU16. The findings of several oncogenes on the same ecDNA with extensive interactions have been shown before - even these same genes in the same cell lines... Figure 4d and Extended Data Fig. 6 from Hung et al (PMID: 34819668) are almost identical to Figure 4 of this manuscript.

- My question on Super-enhancers should be addressed in the manuscript. How many SEs are enhancers for multiple genes? To clarify, I am not questioning the validity of including SEs, just a better understanding of the SE contribution to the HAPI genes.

Reviewer #2 (Remarks to the Author):

The authors have done a good job of addressing the key concerns, particularly with regard to visual confirmation and including copy number information. The paper will be of considerable interest to readers.

REVIEWERS' COMMENTS

Reviewer #1 (Remarks to the Author):

We thank the additional comments from Reviewer #1 regarding our manuscript. We have addressed the comments by modifying the manuscript and performing additional analyses, as shown below:

1. The authors have responded to some of my concerns with additional analysis and explanation, however, it is challenging to identify the changes added without any line- or page numbers or referencing in the rebuttal.

We have now added line and page numbers in the newly revised version.

2. The section of essentially re-discovering ETV1 is still mostly incremental. Finding enhancer regions required for the activation is worthwhile to report, but overall, this section does not add much novelty and should be toned down.

In the revised manuscript, we have made clear that the *ETV1* enhancer hijacking has been reported previously and it is mainly used as a positive control for experimental validation of the HAPI analysis. In addition, we have shortened the *ETV1* part to down tone this finding. Finally, we augmented previous discoveries by showing these enhancers are androgen-receptor driven.

3. A new analysis finding what appears to be competition between promoters on ecDNAs is interesting and novel and should be highlighted. In my view, this is the most interesting part of the revised manuscript. This is not a new mechanism IMO, but an interesting observation that supports the notion that E:P interactions can compete and is zero-sum (on ecDNA).

We appreciate the reviewer's positive comments regarding our finding of the *MYC-ERBB2* ecDNA. Based on the reviewer's suggestion, we have now highlighted this in our Abstract: "We characterized a *MYC-ERBB2* chimeric ecDNA, in which *ERBB2* heavily hijacks *MYC*'s enhancers. Notably, CRISPRi of the *MYC* promoter led to increased interaction of *ERBB2* with *MYC* enhancers and elevated *ERBB2* expression."

4. There is no compelling evidence for a new mechanism (called "mutualistic"). ecDNAs have been shown multiple times to have relaxed chromatin and highly interacting structures. The authors need to remove statements claiming a novel mechanism here.

Based on the reviewer's suggestion, we have now removed the statements claiming mutualistic ecDNA as a new mechanism in our revised manuscript.

5. The authors highlight two *MYC* containing ecDNAs - one in COLO320 and one in SNU16. The findings of several oncogenes on the same ecDNA with extensive interactions have been shown before - even these same genes in the same cell lines... Figure 4d and Extended Data Fig. 6 from Hung et al (PMID: 34819668) are almost identical to Figure 4 of this manuscript.

We cited the previously published COLO320DM and SNU16 data in describing Figure 4 and analyzed this previous data to show that our newly developed HAPI analysis is able to identify enhancer hijacking events on ecDNAs that harbor multiple oncogenes. They are important positive controls to show the robustness of our methodology. We have clarified this point in the section "HAPI analysis identifies enhancer hijacking associated with known ecDNAs" of the revised manuscript. The figures that the reviewer referred to are re-analysis of HiChIP data published by Hung et al., which we have made clear in the figure legends. This is important to show that we were able to recapitulate the enhancer-promoter interactions reported by the previous work before we applied the HAPI analysis to these data.

6. My question on Super-enhancers should be addressed in the manuscript. How many SEs are enhancers for multiple genes? To clarify, I am not questioning the validity of including SEs, just a better understanding of the SE contribution to the HAPI genes.

Based on the reviewer's suggestion, we have now provided additional analyses to assess the number of super-enhancers that are connected to multiple genes (data now presented in Figure S1D). Focusing on super-enhancers that harbor at least one enhancer anchor of E-P loops in LNCaP cells (n=610), we found the majority of the super-enhancers are looped to both HAPI and non-HAPI genes (369), or multiple non-HAPI genes (166). These data are now described in page 3, line 73: "Conversely, super-enhancers are not necessarily looped to HAPI genes in LNCaP cells, as most of the super enhancers interact with both HAPI and non-HAPI genes, or only non-HAPI genes (Figure S1D)."

Figure S1D: The percentage of super-enhancers, involving enhancer-promoter (E-P) loops, that are looped to HAPI and/or non-HAPI genes in LNCaP cells.

Reviewer #2 (Remarks to the Author):

The authors have done a good job of addressing the key concerns, particularly with regard to visual confirmation and including copy number information. The paper will be of considerable interest to readers.

We thank the reviewer's positive comments of our revised manuscript.